



# Effective coefficient of diffusion and permeability of firn at Dome C and Lock In, Antarctica - Estimates over the 100-850 kg m$^{-3}$ density range

Neige Calonne[1], Alexis Burr[2,3], Armelle Philip[2], Frédéric Flin[1], and Christian Geindreau[4]

[1]Univ. Grenoble Alpes, Université de Toulouse, Météo-France, CNRS, CNRM, Centre d'Études de la Neige, Grenoble, France
[2]Univ. Grenoble Alpes, CNRS, IRD, Grenoble INP, IGE, Grenoble, France
[3]Univ. Grenoble Alpes, CNRS, Grenoble INP, SIMaP, Grenoble, France
[4]Univ. Grenoble Alpes, CNRS, Grenoble INP, 3SR, Grenoble, France

**Correspondence:** Neige Calonne (neige.calonne@meteo.fr)

**Abstract.** Modeling air transport through the entire ice sheet column is needed to interpret climate archives. To this end, different regressions were proposed to estimate the effective coefficient of diffusion and permeability of firn. Such regressions are often valid for specific depth or porosity ranges and were little evaluated as data of these properties are scarce. To contribute with a new dataset, this study presents the effective coefficient of diffusion and the permeability at Dome C and Lock In,

Antarctica, from the near-surface to the close-off (23 to 133 m depth). Also, microstructure is characterized based on density, specific surface area, closed porosity ratio, connectivity index and structural anisotropy through the correlation lengths. All properties were estimated based on pore-scale computations on 3D tomographic images of firn samples. Normalized diffusion coefficient ranges from $1.9 \times 10^{-1}$ to $8.3 \times 10^{-5}$ and permeability ranges from $1.2 \times 10^{-9}$ to $1.1 \times 10^{-12}$ m$^2$, for densities between 565 and 888 kg m$^{-3}$. No or little anisotropy is reported. Next, we investigate the relationship of the transport properties

with density over the firn density range as well as over the entire density range encountered in ice sheets by including snow data. Classical analytical models and regressions from literature are evaluated. For firn (550 - 850 kg m$^{-3}$), good agreements are found for permeability and diffusion coefficient with the regressions based on the open porosity of Freitag et al. (2002) and Adolph and Albert (2014), despite the rather different site conditions (Greenland). Over the entire 100 - 850 kg m$^{-3}$ density range, permeability is accurately reproduced by the Carman-Kozeny and Self-Consistent (spherical bi-composite) model when

expressed in terms of a rescaled porosity $\phi_{res} = (\phi - \phi_{off})/(1 - \phi_{off})$ to account for pore closure, with $\phi_{off}$ the close-off porosity. For the normalized diffusion coefficient, none of the evaluated formulas were satisfactory so we propose a new regression based on the rescaled porosity that reads $D/D^{air} = (\phi_{res})^{1.61}$.

## 1   Introduction

Atmospheric air circulates through the interconnected pores of snow and firn down to the firn-ice transition where pores close.

Air entrapped in the closed pores of ice preserved past atmospheric air, from couple of thousands to few millions of years old, providing invaluable data on past Earth's environment (e.g. Petit et al., 1999; Lüthi et al., 2008; Loulergue et al., 2008;



Yan et al., 2019). As gas transport from the surface is slow, air in the open pores of firn can be as old as several decades up to thousand of years (Schwander et al., 1988; Battle et al., 1996; Kaspers et al., 2004). Among others challenges, interpreting firn and ice data with respect to past Earth's conditions and events requires modeling of the air transport processes through

the entire snow-firn-ice column (e.g. Trudinger et al., 1997; Rommelaere et al., 1997; Goujon et al., 2003; Severinghaus and Battle, 2006; Hörhold, 2006; Courville et al., 2007; Witrant et al., 2012; Buizert et al., 2012; Stevens et al., 2020). Two of the important properties for gas transport in snow and firn are the effective coefficient of diffusion and the intrinsic permeability.

The effective diffusion coefficient tensor $\mathbf{D}$ ($\mathrm{m^2\,s^{-1}}$) describes the molecular diffusion of a given gas through a layer of snow or firn, which is a dominant transport process taking place throughout the snow-firn column til the close-off (Schwander

and Stauffer, 1984; Sowers et al., 1992). Defined in a tensorial way, the diffusion coefficient links the gas density gradient $\boldsymbol{\nabla}\rho_g$ ($\mathrm{kg\,m^{-3}\,m^{-1}}$) and the diffusion flux per unit area $\mathbf{J}$ ($\mathrm{kg\,m^{-2}\,s^{-1}}$) through the Fick's law $\mathbf{J} = -\mathbf{D}\boldsymbol{\nabla}\rho_g$. The intrinsic permeability tensor $\mathbf{K}$ ($\mathrm{m^2}$) controls air advection through snow or firn forced by air pressure differences, typically caused by wind at the surface (windpumping) (Colbeck, 1989; Waddington et al., 1996; Kawamura et al., 2006). Permeability links the air pressure gradient $\boldsymbol{\nabla}p$ ($\mathrm{Pa\,m^{-1}}$) and the discharge per unit area $\mathbf{q}$ ($\mathrm{m\,s^{-1}}$) through the Darcy's law $\mathbf{q} = -(1/\mu)\mathbf{K}\boldsymbol{\nabla}p$,

where $\mu$ is the dynamic viscosity of the fluid ($\mathrm{kg\,m^{-1}\,s^{-1}}$). In contrast to molecular diffusion, air advection is not always present in snow and firn and, if so, affects mostly their uppermost meters (Albert, 1996; Albert and Shultz, 2002; Albert et al., 2004). Effective diffusion coefficient and permeability depend on density and open porosity at first order, but also on other microstructural parameters of snow and firn such as pore morphology.

The effective coefficient of diffusion and the permeability of snow and firn were investigated at numerous sites of ice sheets

and glaciers (e.g. Schwander, 1989; Fabre et al., 2000; Albert et al., 2000; Albert and Shultz, 2002; Freitag et al., 2002; Goujon et al., 2003; Rick and Albert, 2004; Hörhold et al., 2009; Courville et al., 2010; Adolph and Albert, 2013, 2014; Sommers et al., 2017). Some come along with a characterization of the microstructure based on 3D images from serial sections (e.g. Rick and Albert, 2004; Freitag et al., 2002) or micro-tomography (e.g. Hörhold et al., 2009; Courville et al., 2010; Adolph and Albert, 2014). Different parameterizations of the properties were suggested (e.g. Schwander et al., 1988; Fabre et al., 2000; Witrant

et al., 2012; Adolph and Albert, 2014), including the ones shown in Table 1. Adolph and Albert (2014) compared different parameterizations of diffusion coefficient and permeability against their measurements at Summit, Greenland. Stevens (2018) compared profiles of effective coefficient of diffusion and $\delta^{15}$N predicted by 6 different parameterizations at NEEM, Greenland and South Pole, Antarctica. Only few parameterizations are based on measurements or modeling over the entire firn column (Adolph and Albert, 2014), limiting their range of validity (Tab. 1). Especially, it is crucial to describe well air transport

properties in the lock-in zone from the beginning of the pore closure to the close-off. These parameterizations require the knowledge of the evolution of the closed porosity with depth (and/or density). Such a prediction is still poorly restricted and only limited parameterizations are available (Schwander, 1989; Goujon et al., 2003; Severinghaus and Battle, 2006; Mitchell et al., 2015; Schaller et al., 2017), which add on uncertainties, as shown by a comparison of parameterizations of closed porosity at Lock In and Vostok, Antarctica by Fourteau et al. (2019). Finally, some conclusions from the above mentioned

studies are that parameterizations of transport properties are strongly site-dependent, which might indicate that regression



based on open porosity or porosity alone is not sufficient and that there is a more complex relationship with microstructure or other environmental parameters (Courville et al., 2007; Adolph and Albert, 2014; Keegan et al., 2019).

In the present study, we provide new datasets of the effective coefficient of diffusion and permeability from the near-surface to the close-off for 2 sites in Antarctica, Dome C and Lock In (plus few data of Vostok). Estimates are based on computations on high-resolution tomographic images of microstructure, as used in many snow studies (e.g. Zermatten et al., 2011; Calonne et al., 2012, 2014b) and in a few firn studies (Freitag et al., 2002; Courville et al., 2010; Fourteau et al., 2019). 3D-image based computations provide the 3D tensor of the properties, allowing to assess the anisotropy of properties and compare lateral to vertical gas transport. A variety of parameter to characterize firn microstructure was also estimated from images. Further, we investigate the relationship of the effective coefficient of diffusion and permeability with density in the firn density range (550 - 850 kg m$^{-3}$), as well as in the entire density range encountered in ice sheet (100 - 850 kg m$^{-3}$) by including data from seasonal snow images from previous studies. Classical analytical models based on simplified microstructures as well as regressions from previous firn studies are evaluated against our results. A new regression is proposed to estimate the diffusion coefficient in the whole density range.

## 2 Methods

### 2.1 3D images

This study is based on a set of 62 3D images of snow, firn or bubbly ice, as used in Calonne et al. (2019). 27 images are samples of firn or bubbly ice from three locations in Antarctica: Dome C, near Concordia Station (75°6'S, 123°21'E), Lock In, located at 136 km away from Concordia Station (74°8.310'S, 126°9.510'E), and Vostok. These samples were extracted from ice cores collected during previous expeditions (Coléou and Barnola, 2001; Gautier et al., 2016; Burr et al., 2018) at depths ranging from 23 to 133 m and show different levels of densification til the close-off. 35 images are samples of snow, covering the main snow types, either collected in the field or obtained from experiments under controlled conditions in cold-laboratory (Coléou et al., 2001; Flin et al., 2004, 2011; Calonne et al., 2014a). The 3D images are binary images (air or ice) of resolutions between 5 and 15 μm and of dimensions between 2.5$^3$ mm$^3$ and 7$^2$ × 25 mm$^3$. Computations of properties were performed on cubic representative elementary volumes of size between 2.5$^3$ and 10$^3$ mm$^3$ for snow and of size 7$^3$ mm$^3$ for firn. More information on the samples and 3D images can be found in Calonne et al. (2019).

### 2.2 Effective coefficient of diffusion and intrinsic permeability

The 3D tensor of the effective coefficient of diffusion **D** (m$^2$ s$^{-1}$) and of the intrinsic permeability **K** (m$^2$) were computed on the set of 3D images of firn. Properties from the images of snow are from the previous studies of Calonne et al. (2012) and Calonne et al. (2014b). A specific boundary value problem, describing vapor diffusion or air flow through the porous medium and arising from a homogenization technique (Auriault et al., 2009; Calonne et al., 2015), is solved on representative elementary volumes of the images using the software Geodict (Thoemen et al., 2008), applying periodic boundary conditions





on the external boundaries. The effective diffusion coefficient was computed with an artificial diffusion coefficient of gas in free air set to $D^{\text{air}} = 1 \text{ m}^2 \text{ s}^{-1}$. In this study, we present the normalized values of the effective diffusion $\mathbf{D}/D^{\text{air}}$ (dimensionless). These normalized values can be multiplied by the diffusion coefficient of the gas of interest in free air to get the physical,

non-normalized values of effective diffusion coefficient of this gas in snow or firn (for example, the diffusion coefficient of vapor in free air, that is $2.036 \times 10^{-5} \text{ m}^2 \text{ s}^{-1}$ at -10°C (Massman, 1998), could be used to get the effective diffusion coefficient of vapor). As the non-diagonal terms of the tensor $\mathbf{D}$ and $\mathbf{K}$ are negligible, we consider only the diagonal terms, i.e. seen as the eigenvalues of the tensors (the image axes $x$, $y$ and $z$ are the principal directions of the microstructure, $z$ being along the direction of gravity). Besides, the tensors are transversely isotropic as the components in $x$ are very similar to the ones in $y$. In

the following, $D$ and $K$ refer to the averages of the diagonal terms of $\mathbf{D}$ and $\mathbf{K}$, respectively. $D_z$ and $K_z$ refer to the vertical components and $D_{xy}$ and $K_{xy}$ refer to the mean horizontal components where $D_{xy} = (D_x + D_y)/2$ and $K_{xy} = (K_x + K_y)/2$. Finally, the anisotropy of the properties is characterized based on the anisotropy ratio $\mathcal{A}(D) = D_z/D_{xy}$ and $\mathcal{A}(K) = K_z/K_{xy}$ (e.g. Calonne et al., 2014a).

## 2.3   Microstructural parameters

The density $\rho$ (kg m$^{-3}$) was computed from 3D images by a standard voxel counting algorithm using an ice density $\rho_i = 917$ kg m$^{-3}$ (ice density variations with temperature were neglected in this study). Porosity corresponds to $\phi = 1 - \rho/\rho_i$ and is the sum of the open porosity $\phi_{\text{op}}$ and of the closed porosity. In the following, density and porosity at the close-off depth are referred as $\rho_{\text{off}}$ and $\phi_{\text{off}}$, respectively. The close-off depth is the depth at which pores are fully isolated from the surface.

  To characterize the connectivity of the pore space, we used the connectivity index (CI) proposed by Burr et al. (2018) defined

as the ratio between the largest pore and the total volume of pores. CI is 100 % when the porosity is fully open and 0 % when pores are closed. This index allows to describe more accurately the pore closure in firn and bubbly ice than the classical closed-to-total porosity ratio, the latter being sensitive to the sample size (Burr et al., 2018). The closed-to-total porosity ratio (CP) is obtained by dividing the total volume of closed pores by the total volume of pores, both estimated on 3D images by counting voxels following (Burr et al., 2018).

The correlation lengths $l_{c_x}$, $l_{c_y}$, and $l_{c_z}$ (mm) were used as a characteristic length of the microstructure in the $x$, $y$, and $z$ direction, respectively. The two-point correlation (a.k.a. covariance) functions for the air phase $S_2(\mathbf{r}_\beta)$ were computed on the 3D images, with $\mathbf{r}_\beta$ a vector oriented along the coordinate axes $\beta = (x, y, z)$ of length $|\mathbf{r}_\beta| = r_\beta$ that ranges from 0 to the image size in the $\beta$ direction with increments of 1 pixel size (Torquato, 2002). The correlation lengths were then determined by fitting the two-point correlations with an exponential equation of form $S_2(r_\beta) = (\phi - \phi^2) \exp(-r_\beta/l_{c_\beta}) + \phi^2$, where $\phi$ is

the porosity (Löwe et al., 2013; Calonne et al., 2014a). The anisotropy ratio $\mathcal{A}(l_c) = l_{c_z}/l_{c_{xy}}$, where $l_{c_{xy}} = (l_{c_x} + l_{c_y})/2$, was used to describe the geometrical anisotropy of the microstructure.

  The specific surface area of snow (SSA) describes the total surface area of the air–ice interface per unit mass (m$^2$ kg$^{-1}$) and was computed from 3-D images using a stereological method (Flin et al., 2011). Providing a characteristic length of the ice grains, the equivalent sphere radius $r$ (mm) is related to SSA by $r = 3/(SSA \times \rho_i)$ (e.g. Grenfell and Warren, 1999; Painter

et al., 2006).



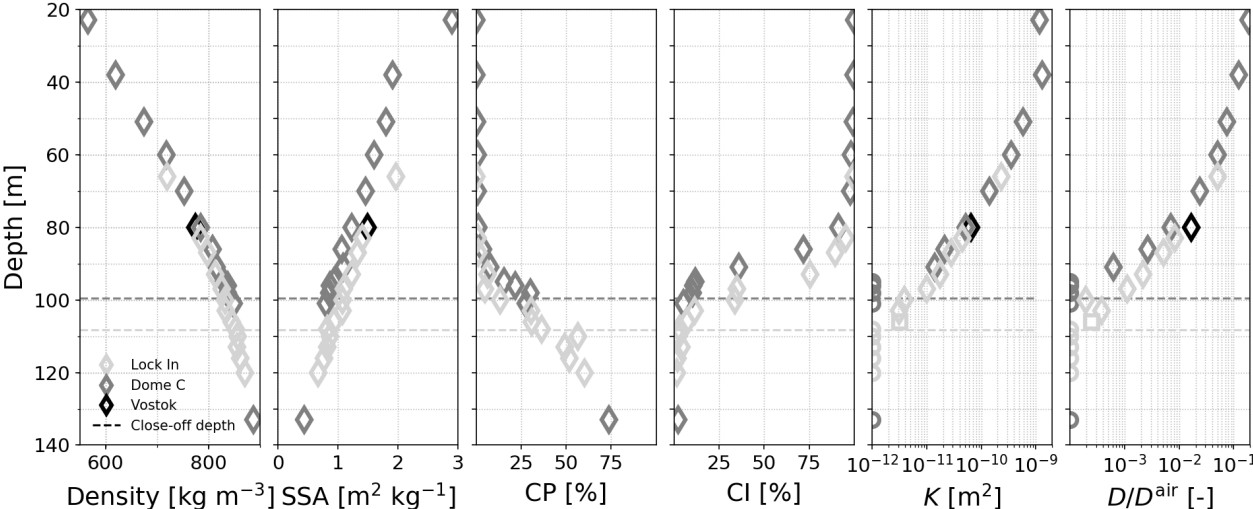

**Figure 1.** Evolution of firn properties with depth: density, specific surface area, closed-to-total porosity ratio, connectivity index, averaged permeability, and averaged effective coefficient of diffusion (from left to right). Dashed lines show the close-off depth at Dome C and Lock In, as given in Burr et al. (2018). Circle symbols indicate values of permeability and effective diffusion equal to zero. Square symbols show a Lock In sample (106 m depth) with horizontal components of the transport properties equal to zero and correspond thus to the values of the vertical component only.

## 3 Results and discussion

### 3.1 Properties at Dome C and Lock In

We present the transport properties and microstructure of firn at Dome C and Lock In (Fig. 1). Firn microstructure is gradually getting denser and coarser with depth: density and SSA evolves from 565 to 888 kg m$^{-3}$ and from 2.89 to 0.43 m$^2$ kg$^{-3}$,

respectively, between 23 and 133 m depth. Whereas the evolution rate is similar at both sites, Lock In shows systematically lower values of density and higher values of SSA compared to Dome C at a given depth. For comparison, the microstructure of the sample from Vostok at 80 m depth is also shown and has a density of 774 kg m$^{-3}$ and a SSA of 1.5 m$^2$ kg$^{-3}$, matching the property profile of Lock In. Air pores start to close from a depth of around 80 m, where the closed-porosity ratio (CP) and the connectivity index (CI) start deviating from the value of 0 and 100 %, respectively. Again, differences in the pore closure

are found between both sites, Dome C showing an earlier onset of pore closure than Lock In. The close-off is thus reached at a depth of 99.3 m at Dome C and, deeper, at 108.3 m at Lock In (Burr et al., 2018). The connectivity index reaches values close to 0 % at these close-off depths and below. In contrast, the closed-to-total porosity ratio keeps on decreasing below the close-off depth, reaching 74% at 133 m at Dome C and 60 % at 120 m at Lock In. A more detailed description of the firn microstructure and pore closure at those sites is provided in Burr et al. (2018).





As the pore space shrinks, transport properties decrease with depth, reaching zero values near the close-off depth. At Dome
C, the averaged values of normalized diffusion coefficient range from $1.9 \times 10^{-1}$ at 23 m depth to $6.2 \times 10^{-4}$ at 91 m depth,
and are equal to zero at 95 m depth and below (5 samples; circle symbols). At Lock In, values range from $5.1 \times 10^{-2}$ at 66 m
depth to $8.3 \times 10^{-5}$ at 106 m depth, and are equal to zero at 108 m and below (5 samples; circle symbols). A coefficient of $1.6$
$\times 10^{-2}$ is found at Vostok at 80 m depth. Regarding permeability, averaged values at Dome C range from $1.2 \times 10^{-9}$ m$^2$ at 23

140    m depth to $1.4 \times 10^{-11}$ m$^2$ at 91 m depth, and are equal to zero below. At Lock In, values range from $2.4 \times 10^{-10}$ m$^2$ at 66
m depth to $1.1 \times 10^{-12}$ m$^2$ at 106 m depth, and are equal to zero below. The Vostok sample shows a value of $6.6 \times 10^{-11}$ m$^2$
at 80 m depth. The small systematic shift in values between Lock In and Dome is also found in the transport properties: Lock
In shows overall higher values of diffusion coefficient and permeability than Dome C for given depths. Finally, relating the
transport properties to the parameters of pore closure in the lock-in zone, we can see that the transport properties reach zero at

or just before the close-off depths (dashed lines in Fig. 1). The zone of zero transport is characterized by connectivity indexes
between 11 and 1 %, reflecting little to no connected pore space. In contrast, the closed-to-total porosity ratio still increases
largely from 15 and 73 %, indicating than even after the close-off, open pores are still present, even down to 133 m depth,
which does not seem consistent with zero transport.

## 3.2   Relationship with density

Next, we study the evolution of the two transport properties with density. Figure 2 includes the simulations on snow samples
to study the relationship over the entire snow-firn density range (102 - 888 kg m$^{-3}$). Figure 3 focuses on firn samples only
(565 - 888 kg m$^{-3}$). Both figures show dimensionless permeability values, i.e. permeability values $K$ divided by the equivalent
sphere radius $r = 3/(\text{SSA} \times \rho_i)$ to the square. This allows to account for the dependency of permeability with a characteristic
length of the microstructure (e.g. Boutin and Geindreau, 2010). The horizontal and vertical component of properties is shown

by the T-shaped symbols.
    Values of the transport properties evolve within several orders of magnitude over the entire density range (Fig. 2). Averaged
values of dimensionless permeability range from 0.9 for the lightest snow sample (PP) to $2 \times 10^{-3}$ for the densest one (MF),
cover the range $10^{-4}$ to $10^{-7}$ for the firn samples, and equal zero-value for the densest firn samples below the close-off. Zero
values are shown for samples of densities above 830 kg m$^{-3}$ at Dome C and for densities above 850 kg m$^{-3}$ at Lock In. For

the diffusion coefficients, averaged values range from 0.75 to 0.17 in snow, from $10^{-2}$ to $10^{-5}$ for firn above the close-off, and
zero values below. Overall, the figures highlight the strong dependency of diffusion and permeability to density (and to SSA for
permeability) with rather well-aligned, little scattered evolution. In contrast to the linear evolution observed for snow (Calonne
et al., 2012), the effective diffusion coefficient of firn shows rather an exponential evolution with density. Both properties see
their values drop when getting near to the close-off density. For example, between 813 and 844 kg m$^{-3}$, the averaged values

of diffusion coefficient drop from $2.2 \times 10^{-3}$ to $8.3 \times 10^{-5}$ and the normalized permeability from $2.4 \times 10^{-6}$ to $8.6 \times 10^{-8}$.
No significant differences are found between sites: they show similar property-density relationship, in contrast with the shift
observed in the property-depth relationship as described above (Fig. 1).



**Figure 2.** Dimensionless permeability and normalized diffusion coefficient of snow and firn versus density: computations on the 3D images ("T" symbols) and analytical models. The proposed regression Eq. 2 is also shown. The sub-caption shows the evolution of the rescaled porosity $\phi_{\mathrm{res}}$ (Eq. 1) compared to the total porosity $\phi$ with density. Snow types correspond to the ICSSG (Fierz et al., 2009): precipitation particles (PP), decomposed and fragmented particles (DF), rounded grains (RG), faceted crystals (FC), depth hoar (DH), and melt forms (MF). The vertical blue lines at 845 kg m$^{-3}$ indicate the close-off density.



**Figure 3.** Dimensionless permeability and normalized diffusion coefficient of firn versus density: computations on the 3D images ("T" symbols) and analytical models (lines). The proposed regression Eq. 2 is also shown. The vertical blue lines at 845 kg m$^{-3}$ indicate the close-off density.





### 3.3 Anisotropy

The anisotropy ratios of both transport properties, $\mathcal{A}(D)$ and $\mathcal{A}(K)$, and their link with the microstructure are presented in
Figure 4. Overall, anisotropy ratios in firn range between 0.8 and 2.3 for the permeability and between 0.04 and 8.2 for the
diffusion coefficient, which correspond to wide ranges compared to the range 0.8 - 1.6 observed for snow. However, looking
at the evolution of the anisotropy ratio with density (Fig. 4a and 4c), we see that the extreme values are found in the narrow
density range 800-840 kg m$^{-3}$, i.e. near the close-off. Those extremes values are reached by dividing very small values of
vertical components over horizontal components of the property tensors (e.g. for Lock In at 103 m depth, $D_{xy}= 1.11 \times 10^{-4}$
and $D_z = 9.18 \times 10^{-4}$, leading to $\mathcal{A}(D) = 8.23$). As these anisotropy ratios concern very small values of properties, they do
not lead to significant impact in terms of gas transport. Most interestingly, in the range 550 - 750 kg m$^{-3}$, anisotropy ratios
$\mathcal{A}(D)$ and $\mathcal{A}(K)$ of firn are comprised between 1 and 1.33 (6 samples). These values are consistent with data of Freitag et al.
(2002), who observed a slight anisotropy of both properties in Summit, Greenland, between 16 and 57 m depth. Finally, when
looking at the evolution at Dome C especially, it seems that anisotropy ratio tends to decrease with depth, in the range 550 -
180 750 kg m$^{-3}$, although more data would be needed for this observation to be significant.

Regarding the geometric anisotropy of the firn samples, the ratios $\mathcal{A}(l_c)$ are rather moderate and do not exceed 0.91 and
1.19, in agreement with anisotropies reported by Burr et al. (2018), and do not show a significant trend with depth or density.
Considering all firn samples, no clear relationship is found for firn between the physical anisotropy $\mathcal{A}(D)$ and $\mathcal{A}(K)$ and the
structural anisotropy $\mathcal{A}(l_c)$. Looking at the 6 firn samples of density below 750 kg m$^{-3}$, a positive correlation can however
be found, following roughly the trend observed in snow, but being less significant given the too few samples. Additional firn
samples in the range 550-750 kg m$^{-3}$ would be needed to fully investigate the evolution of anisotropy ratios with depth and
their relationships.

### 3.4 Comparison to models

Here we evaluate two common models based on simplified microstructures against our data: the self-consistent model for bi-
190 composite spherical inclusions (SC$_{bi}$) and the Carman-Kozeny model (CK), as described in Table 1. In the SC$_{bi}$ scheme, the
medium consists of a bi-composite spherical pattern made of an internal spherical grain and an external fluid shell that ensures
fluid connectivity whatever the porosity value (Boutin, 2000). The SC$_{bi}$ scheme can be used to provide estimates of effective
diffusion coefficient (SC$_{bi}^D$) and estimates of permeability (SC$_{bi}^K$). The Carman-Kozeny model provides permeability estimates
by describing the medium as a bundle of capillarity tubes of equal length (Bear, 1972). Also, for comparison, we show the
195 formulas that provided the best agreements with the snow data (Calonne et al., 2012, 2014b): the self-consistent model of
diffusion coefficient (SC), which is based on a assemblage of spherical particles of air embedded in a homogeneous equivalent
medium whose effective diffusion is the unknown to be calculated (Auriault et al., 2009), and the parameterization of snow
permeability (Calonne 2012). These two formulas are however not suited for firn. All the above mentioned models require the
knowledge of density and, for permeability, of a characteristic length of the microstructure, taken here as the equivalent sphere
radius $r$ of ice grains.


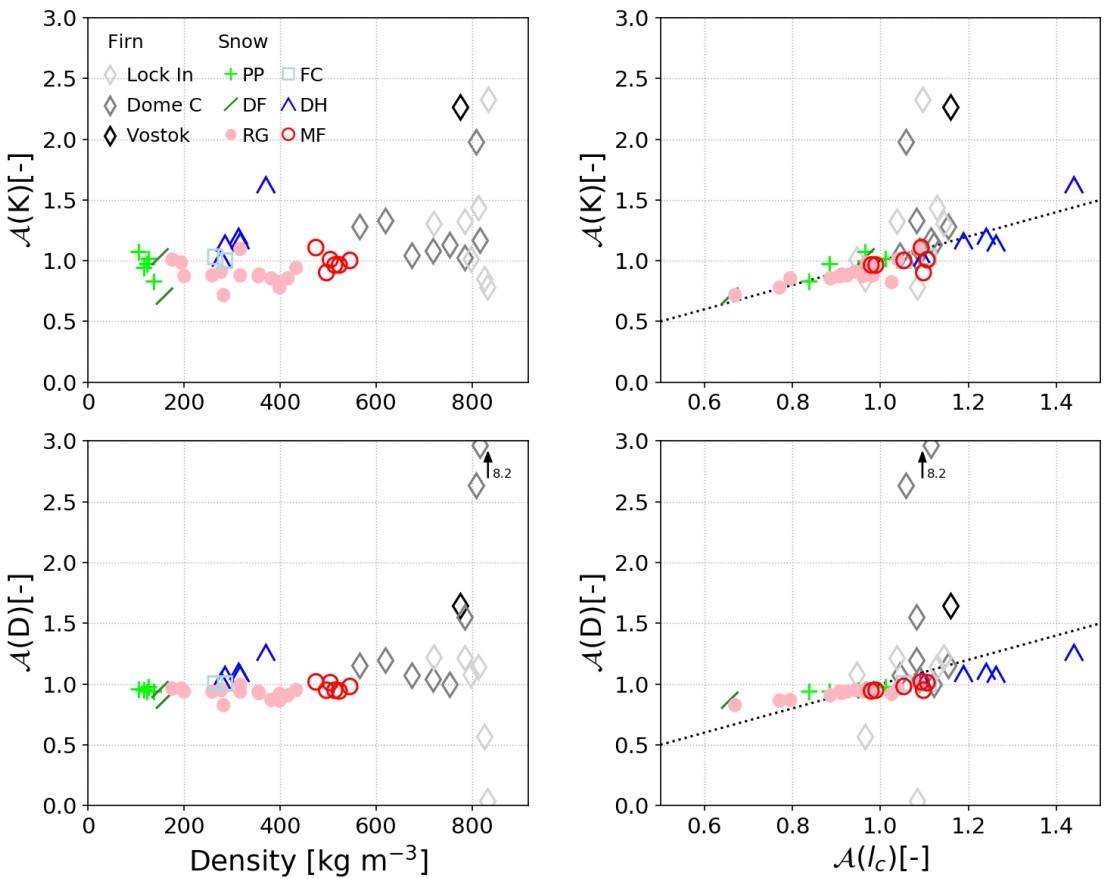

**Figure 4.** Relationships of the anisotropy ratio of the permeability tensor $\mathcal{A}(K)$ (top) and of the diffusion coefficient tensor $\mathcal{A}(D)$ (bottom) with density and structural anisotropy $\mathcal{A}(l_c)$. The arrows indicate a value of $\mathcal{A}(D)$ of 8.2 at a density 832 kg m$^{-3}$ for a firn sample from Lock In.

Model estimates as a function of density are shown in Figure 2 and 3. Taking the models in their original forms (solid lines), none of them succeed to reproduce permeability and effective diffusion coefficient throughout the density range, as they perform badly in firn. This is due to the fact that the models assume that the entire pore space corresponds to open porosity. Zero values of both properties are consequently reached when the porosity is null. To account for pore closure and the fact

that a portion of pores is actually not accessible to gas transport, we introduce a parameter $\phi_{\text{res}}$ that corresponds to a rescaled porosity, defined such as $\phi_{\text{res}} = 0$ at the close-off porosity $\phi = \phi_{\text{off}}$ and $\phi_{\text{res}} = 1$ at $\phi = 1$, and reads

$$\phi_{\text{res}} = (\phi - \phi_{\text{off}})/(1 - \phi_{\text{off}}) \tag{1}$$

A close-off density value of $\rho_{\text{off}} = 845$ kg m$^{-3}$ (close-off porosity $\phi_{\text{off}}$ of 0.078) was taken for all sites, based on our connectivity parameters (Fig. 5). Evolution of $\phi_{\text{res}}$ with density is shown in the sub-figure of Figure 2. $\phi_{\text{res}}$ equals 0.89 at a porosity





| Name | Formula | Validity range; Comments |
|---|---|---|
| **Permeability** | | |
| CK | $K = (4r^2 \times \phi^3)/(180(1-\phi)^2)$ | $0 < \phi < 1$; Carman-Kozeny estimates, (Bear, 1972). |
| $SC_{bi}^K$ | $K = r^2/(3\beta^2) \times [-1 + (2+3\beta^5)/(\beta(3+2\beta^5))]$ with $\beta = (1-\phi)^{1/3}$ | $0 < \phi < 1$; Self-Consistent estimates for bi-composite spherical inclusions (Boutin, 2000). |
| Freitag 2002 | $K = 10^{-7.7} m^2 \phi_{op}^{3.4}$ with $m = 1.5$ | $0.04 < \phi < 0.5$; from pore-scale simulations on samples from 16 to 57 m depth in North Greenland (Freitag et al., 2002). |
| Adolph 2014 | $K = 10^{-7.29} m^2 \phi_{op}^{3.71}$ with $m = 1.5$ | $0.07 < \phi < 0.62$; from measurements on samples from the top 85 m depth at Summit (Greenland) (Adolph and Albert, 2014) |
| Calonne 2012 | $K = 3r^2 \exp(-0.013\rho)$ | $0.4 < \phi < 0.9$; from pore-scale simulations on seasonal snow samples (Calonne et al., 2012). |
| **Normalized diffusion coefficient** | | |
| SC | $D/D^{air} = (3\phi - 1)/2$ | $1/3 < \phi$; Self-Consistent estimates for spherical inclusions (Auriault et al., 2009). |
| $SC_{bi}^D$ | $D/D^{air} = 2\phi/(3-\phi)$ | $0 < \phi < 1$; Self-Consistent estimates for bi-composite spherical inclusions (Boutin, 2000). |
| Eq. (2) | $D/D^{air} = ((\phi - \phi_{off})/(1 - \phi_{off}))^{1.61}$ | $0 < \phi < 1$; from pore-scale simulations of this study. |
| Schwander 1988 | $D/D^{air} = 1.7 \times \phi_{op} - 0.2$ | $0.13 < \phi < 0.5$; from measurements on samples from 2 to 64 m depth at Siple (Antarctica) (Schwander et al., 1988). |
| Fabre 2000 | $D/D^{air} = 1.92 \times \phi_{op} - 0.23$ | $0.15 < \phi < 0.4$; from measurements on samples from Col du Dome (French Alps) and Vostok (Antarctica) (Fabre et al., 2000). |
| Freitag 2002 | $D/D^{air} = \phi_{op}^{2.1}$ | $0.04 < \phi < 0.5$; from pore-scale simulations on samples at 16, 45 and 57 m depth at North Greenland (Freitag et al., 2002). |
| Adolph 2014 | $D/D^{air} = \phi_{op}^{1.5}$ | $0.07 < \phi < 0.62$; from measurements on samples from the top 85 m depth at Summit (Greenland) (Adolph and Albert, 2014). |
| Fourteau 2019 | $D/D^{air} = \phi_{op}^{2.9}$ | $\phi < 0.2$; from pore-scale simulations on samples from about 80 to 110 m depth at Lock In (Antarctica) (Fourteau et al., 2019). |

**Table 1.** Description of the analytical models and regressions of permeability and normalized diffusion coefficient for comparison with our datasets.

of 0.9 (90 kg m$^{-3}$), 0.24 at a porosity of 0.3 (640 kg m$^{-3}$), and 0.02 at a porosity of 0.1 (825 kg m$^{-3}$). Accounting for pore closure through $\phi_{res}$ instead of the open porosity, as classically done for firn property predictive formulas, it is not necessary to introduce an additional relationship to estimate the open porosity from the total porosity, as the one by Schwander (1989) (Eq. 3) for example. Classical models developed for porous media, which do not include open porosity, can then be used for firn by simply replacing the total porosity by the proposed rescaled one.

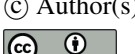



| Name | R2 [-] | MAE [$m^2$ or -] | RMSE [$m^2$ or -] |
|---|---|---|---|
| **Permeability** | | | |
| *over the entire density range* | | | |
| $SC_{bi,res}^{K}$ | 0.78 | $4.60 \times 10^{-10}$ | $8.05 \times 10^{-10}$ |
| $CK_{res}$ | 0.91 | $2.74 \times 10^{-10}$ | $5.23 \times 10^{-10}$ |
| *over the firn density range* | | | |
| $SC_{bi,res}^{K}$ | -0.34 | $1.40 \times 10^{-10}$ | $3.86 \times 10^{-10}$ |
| $CK_{res}$ | 0.17 | $1.08 \times 10^{-10}$ | $3.04 \times 10^{-10}$ |
| Freitag 2002 | 0.87 | $0.43 \times 10^{-10}$ | $1.23 \times 10^{-10}$ |
| Adolph 2014 | -0.58 | $1.22 \times 10^{-10}$ | $4.20 \times 10^{-10}$ |
| **Normalized diffusion coefficient** | | | |
| *over the entire density range* | | | |
| $SC_{bi,res}^{D}$ | 0.96 | $4.3 \times 10^{-2}$ | $5.1 \times 10^{-2}$ |
| Eq. (2) | 0.99 | $1.43 \times 10^{-2}$ | $2.2 \times 10^{-2}$ |
| *over the firn density range* | | | |
| $SC_{bi,res}^{D}$ | 0.37 | $2.7 \times 10^{-2}$ | $3.5 \times 10^{-2}$ |
| Eq. (2) | 0.99 | $0.3 \times 10^{-2}$ | $0.5 \times 10^{-2}$ |
| Freitag 2002 | 0.91 | $0.7 \times 10^{-2}$ | $1.3 \times 10^{-2}$ |
| Adolph 2014 | 0.41 | $2.6 \times 10^{-2}$ | $3.4 \times 10^{-2}$ |
| Fourteau 2019 | 0.42 | $1.4 \times 10^{-2}$ | $3.3 \times 10^{-2}$ |
| Schwander 1988 | -10.1 | $12 \times 10^{-2}$ | $15 \times 10^{-2}$ |
| Fabre 2000 | -14.0 | $14 \times 10^{-2}$ | $17 \times 10^{-2}$ |

**Table 2.** Correlation of determination R2, mean absolute error MAE, and root mean squared error RMSE values obtained between predicted values from models or regressions and true values, being the data computed on 3D images, for permeability and effective coefficient of diffusion, obtained when considering all samples and firn samples only. MAE and RMSE are indicators of the average difference of the modeled values with respect to the true data. RMSE has a higher weight on outliers than MAE, which treats all differences equally. In addition to the average difference between modeled and true data, R2 also indicates how well the model describes the trend seen in the true data (R2 values close to 1 indicate that average differences between model and truth are small and that the true data are equally scattered around the model across the full range).

The CK and $SC_{bi}$ models were modified such as the rescaled porosity $\phi_{res}$ replaces the porosity term $\phi$ in the formulas. These models refer to $CK_{res}$ and $SC_{bi,res}$ in the following. Results are showed by dashed lines in Figure 2 and 3. This modification significantly improves the modeling of effective diffusion coefficient and permeability, especially in the pore closure zone. To quantify the model performance, Table 2 presents the coefficient of determination R2, the mean absolute error MAE, and the root mean squared error RMSE. Permeability is overall well described by the $CK_{res}$ and $SC_{bi,res}^{K}$ model, throughout the density




range (over 9 decades), with MAE and RMSE comprised between 2 and 8 $\times 10^{-10}$ m$^2$. Looking in more details, the CK$_{\mathrm{res}}$
model performs slightly better, being slightly closer to our data especially for light snow below 200 kg m$^{-3}$ (R2 of 0.9 for
the CK$_{\mathrm{res}}$ estimates against 0.78 for the SC$^{\mathrm{K}}_{\mathrm{bi,res}}$ estimates). For the diffusion coefficient, even with the proposed adjustment,
the SC$^{\mathrm{D}}_{\mathrm{bi,res}}$ model overestimates values throughout the density range and especially for the higher densities (R2= 0.96 for all
samples and 0.38 for firn samples).

To provide a satisfactory estimates of diffusion coefficient, we applied a regression of the form $((\phi-\phi_{\mathrm{off}})/(1-\phi_{\mathrm{off}}))^a = \phi^a_{\mathrm{res}}$
to our entire dataset of snow and firn. Here again, we used the proposed rescaled porosity to account for pore closure, and not
the open porosity as many previous regressions (see Sec. 3.5). We obtained the following regression:

$$D/D^{\mathrm{air}} = ((\phi - \phi_{\mathrm{off}})/(1 - \phi_{\mathrm{off}}))^{1.61} = (\phi_{\mathrm{res}})^{1.61} \tag{2}$$

This regression, shown in red lines in Figure 2 and 3, provides estimates with a MAE of 0.014 in the entire density and of
0.0027 in firn (R2 = 0.99 for all samples and for firn samples).

## 3.5 Comparison to regressions from previous firn studies

Figure 5 allows comparing the computed data at Dome C and Lock In with regressions from the studies of Fabre et al. (2000),
Freitag et al. (2002), Adolph and Albert (2014), and Fourteau et al. (2019), as described in Table 1. These regressions were
derived from measurements or pore-scale simulations on firn samples from Antarctica and Greenland, as well as on a few
Alpine specimens. They are all based on the open porosity, which we estimated here with the commonly-used regression of
Schwander (1989):

$$\phi_{op} = \phi(1 - \exp[75(\rho/\rho_{\mathrm{off}} - 1)]) \tag{3}$$

taking $\rho_{\mathrm{off}} = 845$ kg m$^{-3}$. The open porosity fraction derived from the open porosity $\phi_{op}$ from regression of Schwander 1989
($\phi_{op}/\phi \times 100$) and from the closed-to-total-porosity ratio (100 - CP) are compared in Figure 5, together with the connectivity
index CI. In the following comparisons, the performance of the evaluated regressions depends also on the quality of the
Schwander regression (Schwander, 1989), as it was used to estimate the required open porosity. Note that, when taking the
open porosity values provided by the computed CP ratio, the regression performances are worse than when taking the regression
of Schwander 1989, as the CP ratio seems to overestimate the fraction of open pore space (Burr et al., 2018), which leads to
poorer performances.

Permeability predicted by the regressions of Freitag 2002 and Adolph 2014 match overall well our data. Errors from those
regressions are comparable to the ones obtained with the tuned CK$_{\mathrm{res}}$ and SC$^{\mathrm{K}}_{\mathrm{bi,res}}$ models (Tab. 2). The regression of Freitag
2002 performs the best, with MAE and RSME values being about half of the ones shown by the models and the other regressions
for firn. Concerning effective coefficient of diffusion, regressions of form $\phi^n_{op}$ reproduce more closely the evolution with density
compared to regressions of form $a\phi_{op} + b$, as the one proposed by Schwander 1988 and Fabre 2000. These latter regressions
overestimate largely the data within 550 and 750 kg m$^{-3}$ (MAE and RSME between 12 and 17 $\times 10^{-2}$); above, they fail to
reproduce the diffusion coefficient drop at the correct density. Here again, the formula of Freitag 2002 performs best with MAE



**Figure 5.** Top and middle: permeability and normalized diffusion coefficient of firn versus density: computations ("T" symbols) and regressions from literature (lines). Bottom: evolution with density of the open porosity fraction based on the closed-to-total porosity fraction CP and based on the open porosity from Schwander 1989, as well as of the connectivity index CI. The vertical blue lines at 845 kg m$^{-3}$ indicate the close-off density.





and RMSE of $0.7 \times 10^{-2}$ and $1.3 \times 10^{-2}$, respectively, performing closely to our proposed regression (R2 = 0.91 versus R2 = 0.99) and better than the $SC_{bi,res}^{D}$ (R2 = 0.91 versus R2 = 0.37), in the firn density range. The regression of Adolph 2014 and Fourteau 2019 reproduce well the general trend of diffusivity-density relationship but overestimate and underestimate the data overall, respectively. A very good match is however found with the regression of Fourteau 2019 in the 800 - 850 kg m$^{-3}$ density range, where the property drop near the close-off is well reproduced. The good agreement in this density range is consistent with the fact that the regression of Fourteau 2019 was derived from data from Lock In, as in this study, for firn of density above 740 kg m$^{-3}$.

## 4   Conclusions

In this study, we present the effective coefficient of diffusion and permeability at Dome C and Lock In, Antarctica, from near-surface to close-off (23 to 133 m depth). Properties were computed on high resolution 3D tomographic images of firn samples collected in the field. Microstructural parameters, including density, specific surface area SSA, correlation length, structural anisotropy, closed-to-total porosity ratio, and connectivity index, were also estimated. The normalized diffusion coefficient ranges from $1.9 \times 10^{-1}$ to $8.3 \times 10^{-5}$ and permeability from $1.2 \times 10^{-9}$ to $1.1 \times 10^{-12}$ m$^2$, decreasing with depth. Density varies between 565 and 888 kg m$^{-3}$ and SSA from 2.9 to 0.4 m$^2$ kg$^{-1}$, from top to bottom of the firn columns. Between both sites, the evolution of transport properties with depth follows a similar trend but is shifted in depth. Effective coefficient of diffusion and permeability are systematically slightly higher at Lock In than Dome C, for given depths. They reach zero value below 95 m at Dome C against 108 m at Lock In. This can be related to differences in firn microstructure between both sites (Burr et al., 2018): denser and coarser firn is found at Dome C for given depths, the onset of pore closure appears earlier at Dome C, and the close-off is reached at 99 m at Dome C and 108 m at Lock In.

The relationship of the transport properties with density was further investigated within the firn density range as well as in the entire 100 - 900 kg m$^{-3}$ range by including simulations on seasonal snow samples. The relationship of permeability with SSA was also accounted by analyzing the dimensionless permeability, i.e. the permeability divided by the equivalent sphere radius to the square. Over the full density range, transport properties evolve within several orders of magnitude, covering $10^{-1}$ to $10^{-7}$ m$^2$ for dimensionless permeability and $10^{-1}$ to $10^{-5}$ for the normalized diffusion coefficient. Little scattered evolution, without variability linked to sites, is reported over the entire density range, highlighting the strong dependency of transport properties with density (and SSA for permeability).

For firn (550 - 917 kg m$^{-3}$), we report very good agreement with the regression of Freitag 2002 and, in a lesser way, with Adolph and Albert (2014) for both diffusion coefficient and permeability, although their dataset originate from different environments (Greenland versus Antarctica here). In the narrow range of 800 to 850 kg m$^{-3}$, near the close-off, the drop of diffusivity with density observed in our data is closely reproduced by the regression from Fourteau 2019 that is based on data from Lock In.

Looking at the entire range of density (100 - 917 kg m$^{-3}$), permeability is overall well predicted by the Carman-Kozeny and the Self-Consistent (spherical bi-composite) models when modified to account for the pore closure. To do so, the total



porosity $\phi$ was simply replaced by a rescaled porosity $\phi_{\mathrm{res}}$ defined as $\phi_{\mathrm{res}} = (\phi - \phi_{\mathrm{off}})/(1 - \phi_{\mathrm{off}})$, with $\phi_{\mathrm{off}}$ the close-off porosity. We specifically choose to account for pore closure through such a rescaled porosity instead of the commonly-used open porosity. The advantages are that (1) there is no need of an additional predictive formula, as required to estimate the open porosity, which limits uncertainties, and (2) any models developed for porous media which do not include open porosity can be used by doing this simple replacement. For the diffusion coefficient, none of the evaluated models or regressions provide

satisfactory estimates over the entire density range. We thus propose a new regression based on the rescaled porosity that reads $D/D^{\mathrm{air}} = (\phi_{\mathrm{res}})^{1.61}$.

    Finally, as polar snow in the range 100 - 550 kg m$^{-3}$ can differ significantly from seasonal snow in the same density range, it would be interesting to analyze polar snow and firn between 0 and 25 m depth, thus complementing the present dataset. Further studies should also be undertaken to derive transport properties at other sites and evaluate the proposed regression and models

in different environments.

*Data availability.* The computed values of effective diffusion coefficient and permeability are available in the supplement of the article, together with the computed microstructural parameters.

*Author contributions.* N. Calonne wrote the paper with input from A. Burr , C. Geindreau, A. Philip, and F. Flin. Data analysis was performed by N. Calonne, A. Burr, C. Geindreau, and A. Philip. 3D image simulations were performed by C. Geindreau. Sample acquisitions were

performed by A. Philip, A. Burr, F. Flin, and N. Calonne. C. Geindreau, A. Philip, and N. Calonne directed the project.

*Competing interests.* The authors have declared that no competing interests are present.

*Acknowledgements.* The 3SR lab is part of the Labex Tec 21 (Investissements d'Avenir, Grant Agreement ANR-11-LABX-0030). CNRM/CEN and IGE are part of Labex OSUG@2020 (Investissements d'Avenir, Grant ANR-10-LABX-0056). Authors acknowledge the Labex CEMAM (Center of Excellence of Multifunctional Architectured Materials, "Investments for the Future" Program, Grant ANR-10-LABX-44-01) for

its contribution for the funding of the X-ray tomography equipment. We also thank the ID19 beamline of the ESRF for the acquisition of several tomographic images used in this study.



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
