# Peer review of "Effective coefficient of diffusion and permeability of firn at Dome C and Lock In, Antarctica, and of various snow types - Estimates over the 100-850 kg m-3 density range"

_The Cryosphere, 2021_

## Author Comment (AC1)

**Author replies to referee comment RC1**

We thank very much Johannes Freitag for his comments that help improving the manuscript. Please find below our point-by-point replies in green color.

General comments:
The study of Calonne et al. provides new estimates of effective coefficients of diffusion and permeability of polar firn from two low accumulation sites in Antarctica (DomeC and LockIn) and from various snow types based on pore-scale computations on X-CT-images of cm-sized samples. New regressions are proposed based on a rescaled porosity. The rescaled porosity is an easy-to measure parameter and substitutes open porosity in former regressions with the implication that the new formula becomes applicable without assumptions/measurements of open porosity.

The study addresses a relevant scientific question within the scope of TC. It contributes to an improved modelling approach of air-transport in firn and relates directly to the interpretation and dating of ice core records. In my opinion the scientific methods are valid but not well outlined: for the modelling part I miss a comprehensible introduction how diffusion and permeability are actually solved on the three-dimensional lattice. Concerning the description of the computations, we followed the same methods presented in details in previous papers. We made it clearer and added a sentence in the new version of the paper that refers to those papers, L94 : « A comprehensive description of the computation method can be found in Calonne et al. (2012) for the permeability and in Calonne et al. (2014a) for the diffusion coefficient." Also, we included the numerical method used in Geodict to compute permeability and diffusion that is the finite difference method (L97). For the CT-data part I could not find any discussion about the limited sample size (7x7x7mm3) and its implications on the estimates of diffusivity / permeability.
Computations were performed on volumes of 6.7 x 6.7 x 6.7 mm$^3$, which was the minimum volume size to obtain representative values of permeability (representative elementary volume REV). We choose the permeability to define the REV as it is the most critical property among the ones computed in the paper, as it requires the largest volume. REVs for permeability were estimated on selected firn samples by computing the property over sub-volumes of increasing sizes. REV was assumed to be reached when the property values do not vary significantly anymore when increasing the volume size. The figures below provides an illustration of the REVs analysis for two Dome C samples at 23 and 80 m depth for permeability and thermal conductivity (for comparison). We clarify this point about the REVs analysis in the paper in Section 2.1.

[Figure]

[Figure]

In the text it was mentioned that the CP-estimates are affected by the limited size, the CI-estimates maybe not. Interestingly, the open porosity profile (Schwander 1989) shows a different behaviour than both, CP and CI estimates as well (figure 5). It would be worth to know the portion of pores cutted at the surfaces. I would suspect that the high (100-CP) values for high densities result from the large amount of cutted pores that are not counted as isolated pores but count for the total pore volume. On the other hand the CI (largest pore/total pore volume) tends to be underestimated at the percolation threshold in small samples because large pores become less frequent and might be not adequate presented in the sample. This would underestimate the transport properties at the percolation threshold (= closeoff area) as well. I would really appreciate a discussion of the pore-cut effect as it is not even mentioned at all.

In this paper, the pore connectivity is described based on two parameters, the closed-to-total porosity ratio (CP), as classically used in the firn community, and the connectivity index (CI), a recently-introduced parameter by Burr et al. 2018. The latter is dedicated to the study and comparison of both parameters at Lock In and Dome C, based on 3D image computation on firn samples. In particular, they performed a comprehensive analysis on the errors due to sample size, voxel resolution, and spatial heterogeneities on the estimates of CP and CI. Figure 5 of the present paper shows the evolution of the open porosity (100 – CP) and the connectivity index CI, as computed on our set of images, as well as the open porosity from the regression of Schwander 1989. As mentioned by the reviewer, in this figure, the (100 - CP) values are higher than the open porosity values from the regression of Schwander 1989. Burr et al. 2018 showed that CP is sensitive to sample size such as small samples lead to underestimate CP; this could thus explain the overestimation of (100 – CP) compared to Schwander 1989 (Fig. 5) and would indicate that our sample size are not large enough. The dependency of CP on sample size is likely due to its method of computation where the portion of cut pores at the volume boundaries – not counted as closed pore – varied with the sample size, as pointed out by Burr et al. 2018 and by the reviewer. Concerning CI, no significant effect of the sample size was reported by Burr et al. 2018 (see figure 6) within), especially when relating CI to density, as presented in Figure 5 of our paper. This does not support the suggestion of the reviewer that CI would be underestimated in small samples. Also, unlike our CP estimates, taking the open porosity from Schwander 1989 as reference for a direct comparison of our CI estimates might not be adequate as they are different parameters per definition. It is not the scope of this paper to provide a comprehensive analysis of CP and CI as it was addressed in great details in Burr et al. 2018 for the same firn samples. However, we agree that additional information on the estimations of CP and CI are required to help the reader interpreting the results. Section 2.3 was modified along this line (L116).

In respect to the conclusions I am wondering if the implications that both investigated sites show no significant differences in the transport property-density relationships can give further insides into the general properties of polar firn. However due to missing information (how much is the difference in accumulation between DomeC and LockIn?, what is the reason for the huge difference in Close off-depth of 10m although the sites are less than 150km apart?) /adequate figures I was not able to really assess if the similarity in transport properties between the sites are expected or not.

We agree with the reviewer and included the main characteristics of the sites that are the mean annual temperature and the mean annual accumulation rate (L79).

The abstract provides a concise summary, the presentation is well structured, the language fluent and precise. Number and quality of references are appropriate. A few minor points are listed below.

Minor points/suggestions:

line0: I would prefer a different title like "Effective coefficient of diffusion and permeability of firn at DomeC and LockIn, Antarctica and various snow types– estimates over the 100-850kgm-3

density range. The present title is a little bit misleading because at DomeC and LockIn there is no 100 kgm-3 snow, even at surface the snow is already of the order of 300 kgm-3. The authors compiled measurements of DomeC/LockIn firn (550kgm-3 and denser, 23m-133m depth) and alpine/artifical snow (100kgm-3 to 550 kgm-3).

We agree with the proposition of the reviewer; it is more accurate to mention the snow samples in the title. The title has been modified as suggested.

Line 1: ..,the entire ice sheet column... -> (change to) ...the entire firn column of polar ice sheets…

The sentence was modified as suggested.

Line 10: ...with density over the firn density range...snow data. -> ..with density by extending the data set with data from alpine and artificial snow.

The sentence was modified as suggested.

Line 23: ...to thousand of years… -> ...to hundreds of years... (the air in the open pores is not too old, it is the age-difference between air and ice that can be reach thousand of years...)

We corrected this mistake.

Lines 72,73: please provide some additional information about the DomeC, LockIn and Vostok site such as mean annual temperature and accumulation rate to better assess the differences in microstructure and transport properties between the cores.

We included the characteristics of the sites as follows: L79: "Mean annual temperature and mean annual accumulation rate are of -55∘C and 2.5 cm $yr^{-1}$ at Dome C, of -53.15∘C and 4.5 cm $yr^{-1}$ at Lock In, and of -57∘C and 2.2 cm $yr^{-1}$ at Vostok (from Burr et al., 2018, and references within).

Line 79: explain the term: "representative elementary volume". In what respect is the volume "representative"?

Computations were performed on volumes equal or larger than the representative elementary volume estimated for permeability, which is the property that requires the largest volume among the other properties computed in the paper. We clarify this point in the paper: L84: "Computations of properties were performed on cubic images of size between $2.5^3$ and $10^3$ $mm^3$ for snow and of size $6.7^3$ $mm^3$ for firn. These image correspond to volumes equal or bigger than the representative elementary volumes estimated for permeability, which is the property that requires the largest volume among the other properties computed in this paper (Kanit et al, 2003, Calonne et al. 2012, Calonne et al. 2014a). Estimations of the representative elementary volumes for permeability were performed on selected images following Calonne et al. 2011 by computing the property over subvolumes of increasing sizes."

Lines 84,85,86: a few more sentences about how the diffusion coefficients and permeabilities are estimated would be helpful to understand the approach solved by software Geodict.

Concerning the description of the computations, we followed the same methods presented in our previous papers. We think it is not needed to present them in the same level of details in the present paper. However, we did not include the references of the method used in the present paper. We thus added the following sentence L94 : « A comprehensive description of the computation method can be found in Calonne et al. (2012) for the permeability and in Calonne et al. (2014a) for the diffusion coefficient.". Also, we included the numerical method used in Geodict to compute permeability and diffusion that is the finite difference method (L97).

Line 105: ...CI is 100% when the porosity is fully open and 0% when pores are closed...

Comment: CI defined as the ratio between the largest pore and the total pore volume can not be 0% per definition. It tends to 1/bubble-number if all pores are closed and approximately of equal volume.

We agree with the comment and modified the sentence as: L117 "CI is 100 % when the porosity is fully open and decreases as pores shrink and close. For bubbly ice where all pores are closed and of approximately equal volume, CI would tend to 1 over the bubble number."

Line 133: maybe you can add an additional sentence about the CI and CP behaviour at 85m depth: here CI is 75% whereas CP is close to 0%. -> possible interpretation: pore- cut effect(?): pores cutted at the surface loose (artificially) the connection to the open pore space within the volume.
At Lock In, at 87 m CP is 1,2% and CI is 89%. At Dome C, at 86m, CP is 3,8% and CI is 72%. Both parameters reflect the onset of pore closure, with CI and CP moving away from 100 % and 0%, respectively.

Lines 146,147: ...,indicating than even after the close-off, open pores are still present, even down to 133 m depth, which ... This is a misleading sentence: I would rather think that the <100% of CP below the close-off depths are caused by the amount of cutting pores that are not counted as isolated pores – again a pore-cut effect.
We agree and modified the sentence as (L163): "In contrast, the closed-to-total porosity ratio still increases largely from 15 and 73%, indicating erroneously the presence of open pores after the close-off and even down to 133 m depth. This underestimation of the closed-to-total porosity ratio might be related to the sample size effect reported by Burr et al. 2018 and might indicate that our volume of computation are too small to allow a correct estimate of the closed-to-total porosity ratio."

In contrast, the closed-to-total porosity ratio still increases largely from 15 and 73 \% and does not reach 100 \%, indicating erroneously the presence of open pores after the close-off and even down to 133 m depth. This underestimation of the closed-to-total porosity ratio certainly comes from the cut-pore effect which is related to the surface to volume ratio and also to the sample size which effect was reported by Burr et al. 2018. It might indicate that the volumes used for computations are too small to allow a correct estimate of the closed-to-total porosity ratio and would require a correction (e.g. Schaller et al. 2017).

Figures 2, 3: What is the size of the "T" symbols in Figures 2,3 telling us? Is it the std of a set of measurements? How many measurements/ or estimates are averaged?
The "T" symbols corresponds to our computation results and allows to assess the anisotropy of the property with the tip of the "T" symbols being the vertical property component and the horizontal bar of the "T" symbols being the horizontal component. The legend and description in the paper of Figure 2 and 3 have been modified to clarify the meaning of the "T" symbols. L175: "The T-shaped symbols in both figures are our computed property values and the tips and horizontal bars indicate the vertical and horizontal component of the property, respectively."

Lines 278,283: ...For firn (550-917kgm-3)... -> For firn (550-850kgm-3)...Samples with density in the range between 850 kgm-3 and 917kgm-3 are referred to as bubbly ice(with isolated bubbles). The sentence was modified as suggested.

---

## Author Comment (AC2)

**Author replies to referee comment RC2**

We thank very much Zoe Courville for her comments that help improving the manuscript. Please find below our point-by-point replies in green color.

The manuscript presents an improved method for defining transport properties in snow and firn through the examination of microstructural parameters derived from micro CT results and numerically derived values of permeability and diffusivity. The authors propose a simple yet elegant concept of rescaled porosity which accounts for the effects of pore closure on transport values in denser/deeper firn. This approach has promise to improve prediction in firn over previous methods using open porosity of the firn. The fact that the approach improves regression models over a wide range of density values at different polar locations with different conditions also suggests the method is a promising step in developing a generalized snow permeability and diffusivity model.

There are some mostly minor technical edits that should be addressed that are listed by line number below. In addition, I have included suggestions to clarify some of the labeling and description of the figures,

Abstract: As written, it is not clear what was done as part of this research and what was past work (need to make the tense of the sentences consistent throughout).
I.e., on line 1, "To this end, different regressions were proposed to estimate the effective coefficient of diffusion and permeability of firn." When I read this, I get the sense that this is part of the work that is being presented. I think it could be written as (just a suggestion), "To this end, different regressions have been proposed in the past to estimate the effective coefficient of diffusion and permeability of firn,"
The sentence was modified as suggested.

Line 3 (and example of shifting tense): "were little evaluated as data of these properties are scarce" could be written, "have not been evaluated very often as data of these properties are scarce"
The sentence was modified as follows: L3: "These regressions are often valid for specific depth or porosity ranges only. Also, they constitute a source of uncertain as evaluations have been limited by the lack of reliable data of firn transport properties."

Line 10: "by including snow data." What snow data does this refer to? The micro-CT data? Or are you referring to surface snow vs. firn at depth?
We refer to a previously published data of diffusion coefficient and permeability for snow samples from Calonne et al. 2019. The sentence was modified to be clearer: L10 "Next, we investigate the relationship of the transport properties with density over the firn density range (550 – 850 kg m$^{-3}$) as well as over the entire density range encountered in ice sheets (100 – 850 kg m$^{-3}$) by extending the datasets with transport properties of alpine and artificial snow from previous studies"

Line 11: "Classical analytical models and regressions from literature are evaluated." Evaluated compared to what?
The sentence was modified as follows: L13: "Classical analytical models and regressions from literature are evaluated against the estimates from pore-scale simulations."

Line 15 "with φoff the close-off porosity." Should be "with φoff equal to the close-off porosity." Or something similar since with "φoff the close-off porosity" is not a complete phrase.
The sentence was modified as follows: L18: "where $\varphi_{off}$ is the close-off porosity."

Line 20: Air entrapped in the closed pores of ice preserved past atmospheric air, from couple of thousands to few millions of years old, providing invaluable data on past Earth's environment. Couple of suggestions: "preserved" should be "is preserved" or "preserves" "from couple of thousands to few millions of years" should be "on the order of a few thousand years to a few million years old" or "thousands to millions of years old"

We agree with the reviewer and modified the sentence as: L22 "Air entrapped in the closed pores of ice preserves past atmospheric air, thousands to millions of years old, providing invaluable data on past Earth's environment".

Line 23: "Among others challenges" should be "Among other challenges"

The correction was made accordingly.

Line 29: "til" should be "until"

The correction was made accordingly.

Line 48: "Only few parameterizations are based on measurements or modeling over the entire firn column (Adolph and Albert, 2014), limiting their range of validity (Tab. 1)." is slightly confusing, suggest rewriting as "Few parameterizations are based on measurements or modeling…"

The correction was made accordingly.

Line 49: "Especially, it is crucial to describe well air transport properties in the lock-in zone from the beginning of the pore closure to the close-off." should be: It is especially crucial to describe air transport properties well in the lock-in zone

The correction was made accordingly.

Line 59: "plus few data of Vostok" should be "in addition to a few data from Vostok"
It would be helpful to explain here what those data are, since it starts to get confusing in the results section and in Figure 1 about what data from Vostok were used. I.e., it seems like it was density, SSA, diffusivity and permeability from 80 m depth. Maybe it makes sense to say that, i.e., "in addition, density, SSA and diffusivity and permeability from 80 m depth from Vostok were used."

We agree with the reviewer and add the following sentence L61:" Properties of a firn sample from 80 m depth at Vostok, Antarctica, are also presented for additional comparisons".

Line 62: "allowing to assess the anisotropy of properties and compare lateral to vertical gas transport"

Line 63: "A variety of parameter to characterize firn microstructure was also estimated from images" should be "A variety of parameters to characterize…"

The sentence was modified as follows: L65 "In addition to transport properties, a variety of parameters to characterize the firn microstructure was computed from the images."

Line 75: til should be until

The correction was made accordingly.

Line 76: "controlled conditions in cold-laboratory" should be "controlled conditions in a cold-laboratory" or r controlled conditions in the cold-laboratory

The correction was made accordingly.

Line 106: "This index allows to describe more accurately the pore closure in firn and bubbly ice than the classical closedto-total porosity ratio, the latter being sensitive to the sample size (Burr et al., 2018)." Should be "This index allows the pore closure in firn and bubbly ice to be more accurately described than…"

The correction was made accordingly.

Figure 1: The circles are hard to see in the figure for diffusivity and permeability because they show up as half circles in the figures and resemble the diamonds. Is there another symbol that can be used to make it clearer? Also, the square symbols are very hard to see. Could they be black instead of grey so that they show up better? I also cannot see the black diamond indicated the Vostok values for CP and CI, but again, this is unclear if this was calculated for the Vostok core. It's also hard, but not impossible, to see theVostok value indicated by the black diamond on the Density graph. Not sure if there is a way to make the black diamond more visible? Can it be layered on top of the other sites' data points?

Last thing, it would be helpful if the two dashed lines that indicate the close-off depths were either listed in the legend (i.e., if the difference between the two shades of grey were designated), or if there was a label on the figure that indicated which dashed line designated the Dome C close-off depth and which line designates the Lock In close-off depth. If that makes the figure too cluttered, perhaps that can at least be specified in the figure legend.

The correction was made accordingly.

Figure 1 was modified to address the reviewer comments. The revised version is shown below. Especially, the Vostok values for CP and CI were missing and were thus added in the new version. The two dashed lines indicating the close-off depth were labeled directly on the figure. The legend was also modified such as "Dashed lines show the close-off depth at Dome C (dark grey) and Lock In (light grey)".

[Figure]

Line 124: evolves should be evolve
The correction was made accordingly.

Line 129: value should be values
The correction was made accordingly.

Line 152: "Both figures show dimensionless permeability values, i.e. permeability values K divided by the equivalent sphere radius r = 3/(SSA × ρi) to the square." I'm not sure what "to the square" refers to. Is it that the squares in the figure denote the dimensionless permeability? That could be written as "as designated by the squares in Figure 2."

By "the equivalent sphere radius to the square" we meant the squared equivalent sphere radius. To clarify, we reformulate the sentence as follows: L156 "Both figures show dimensionless permeability values, i.e. permeability values K divided by the squared equivalent sphere radius $r^2=$ $(3/(SSA\times\rho i))^2$".

Figure 2: What do the T symbols pointing up or down indicate? Are the in different directions just so that they show up differentiated from one another when they overlap, or is there a physical significance? Also, should the rescaled porosity be defined earlier in the paper so that the inset in the figure is better defined? As it is, the definition doesn't come for another couple pages.

The "T" symbols corresponds to our computation results and allows to assess the anisotropy of the property with the tip of the "T" symbols being the vertical property component and the horizontal bar of the "T" symbols being the horizontal component. The legend and description in the paper of Figure 2 have been modified to clarify the meaning of the "T" symbols. L175: "The T-shaped symbols in both figures are our computed property values and the tips and horizontal bars indicate the vertical and horizontal component of the property, respectively."

Line 173: "Those extremes values" should be "extreme values"
The correction was made accordingly.

Line 196: "a assemblage" should be "an assemblage"
The correction was made accordingly.

Figure 4 caption: What do the dotted lines represent in the right-hand figures?
We included the following sentence in the caption of Fig 4: "Dotted lines indicate 1:1 lines."

Table 1: This table is a very nice summary of the available relationships for permeability and diffusivity.

Line 225: "To provide a satisfactory estimates of diffusion coefficient" should be "To provide satisfactory estimates of diffusion coefficient"
The correction was made accordingly.

Line 232: "Figure 5 allows comparing the computed data" should be "Figure 5 compares the computed data"
The correction was made accordingly.

Line 245: "Permeability predicted by the regressions of Freitag 2002 and Adolph 2014 match overall well our data." should be "Overall, permeability predicted by the regressions of Freitag 2002 and Adolph 2014 match our data well."
The correction was made accordingly.

Line 279: " although their dataset originate" should be "although their datasets originate"
The correction was made accordingly.